# At the frontlines of digitisation: a qualitative study on the challenges and opportunities in maintaining accurate, complete and timely digital health records in India's government health system

Kerry Scott ![ORCID],[1] Osama Ummer ![ORCID],[2] Sara Chamberlain ![ORCID],[3] Manjula Sharma,[4] Dipanwita Gharai,[2] Bibha Mishra,[2] Namrata Choudhury ![ORCID],[5] Diwakar Mohan,[1] Amnesty Elizabeth LeFevre ![ORCID] [6]

For numbered affiliations see end of article.

**Correspondence to**
Dr Kerry Scott;
kscott26@jhu.edu

## ABSTRACT

**Objectives** To understand factors underpinning the accuracy and timeliness of mobile phone numbers and other health information captured in India's government registry for pregnant and postpartum women. Accurate and timely registration of mobile phone numbers is necessary for beneficiaries to receive mobile health services.

**Setting** Madhya Pradesh and Rajasthan states in India at the community, clinical, and administrative levels of the health system.

**Participants** Interviews (n=59) with frontline health workers (FLHWs), data entry operators, and higher level officials. Focus group discussions (n=12) with pregnant women to discuss experiences with sharing data in the health system. Observations (n=9) of the process of digitization and of interactions between stakeholders for data collection.

**Primary and secondary outcome measures** Thematic analysis identified how key actors experienced the data collection and digitisation process, reasons for late or inaccurate data, and mechanisms that can bolster timeliness and accuracy.

**Results** Pregnant women were comfortable sharing mobile numbers with health workers, but many were unaware that their data moved beyond their FLHW. FLHWs valued knowing up-to-date beneficiary mobile numbers, but felt little incentive to ensure accuracy in the digital record system. Delays in registering pregnant women in the online portal were attributed to slow movement of paper records into the digital system and difficulties in gathering required documents from beneficiaries. Data, including women's phone numbers, were handwritten and copied multiple times by beneficiaries and health workers with variable literacy. Supervision tended to focus on completeness rather than accuracy. Health system actors noted challenges with the digital system but valued the broader project of digitisation.

**Conclusions** Increased focus on training, supportive supervision, and user-friendly data processes that

## Strengths and limitations of this study

► This study was strengthened by the use of multiple research methods (interviews, focus groups, and observation) and engagement of multiple frontline stakeholders: the women whose data are entered into the system, the frontline health workers who care for them and create the initial (paper) records, the data entry operators who digitise these records, and the managers, administrators, and leaders who supervise these processes.

► Frontline health workers and data entry operators were aware that their work was being observed and, thus, may have minimised or hidden unauthorised behaviour, such as shortcuts to speed up data entry or unsanctioned task shifting.

► Data collection occurred just prior to the launch of a new digitisation strategy in some (but not all) regions, wherein frontline health workers would be given tablets and asked to directly digitise patient health data; thus, some of the challenges identified in this study may have self-resolved due to this new (shorter) dataflow pathway; and new challenges to accuracy, timeliness, and completeness may have arisen.

prioritise accuracy and timeliness should be considered. These inputs can build on existing positive patient–provider relationships and health system actors' enthusiasm for digitisation.

## INTRODUCTION

Health information systems (HIS) capture data on patients and their contacts with routine health services. Data collected about patients include clinical content, such as weight and blood pressure, and non-clinical

content, such as their address, patient identification code and phone number. HIS support responsive policymaking, resource allocation, routine service delivery and health systems accountability.[1] The achievement of these functions is largely dependent on the accuracy, completeness and timeliness of the data in HIS.[2] Information communication technologies (ICTs) are increasingly being used to improve the core functions of HIS through digitisation, wherein patient data are entered into a computer system rather than maintained in handwritten records.[3–5] ICTs for HIS have great potential to strengthen health systems by streamlining data collection and entry, accelerating the transmission and analysis of data, instituting validity checks and increasing frontline worker access to a range of clinical and administrative support services.[6–8]

The digitisation of HIS has occurred concurrently with increasing mobile phone penetration at a population level and, in turn, an increase in mobile health (mHealth) initiatives. mHealth programmes can transmit health-related information, send reminders and solicit patient feedback on services.[9] To attain high population level coverage at scale, mHealth programmes often rely on beneficiary mobile numbers and other clinical and non-clinical data that have been collected through government HIS. The accuracy, completeness and timeliness of records in government systems can directly affect the capacity of mHealth programmes to reach target populations at the right time points.

The Government of India's Kilkari programme is one example of an mHealth service that relies on the quality and timeliness of mobile phone numbers and other data collected in the government's health registries in order to reach beneficiaries. Developed and scaled by BBC Media Action in collaboration with the Indian Ministry of Health and Family Welfare, Kilkari is the world's largest maternal mobile messaging initiative, having reached 10 million subscribers in 13 states by December 2018.[10] It delivers maternal, child and reproductive health information content through up to 72 once-weekly outgoing prerecorded calls. Beneficiaries (pregnant or postpartum women and their husbands)[11] are subscribed to Kilkari based on the mobile number captured in governmental HIS registries called, depending on the state, the Maternal and Child Health Tracking System (MCTS) or Reproductive and Child Health (RCH) system.[10] The gestational age estimate or child's date of birth is used to determine the appropriate Kilkari messages to send based on the stage in pregnancy or child's age.

Missing and incorrect data in MCTS/RCH reduce the number of beneficiaries who can be exposed to Kilkari. In 2018, one quarter of the unique mobile numbers registered for Kilkari were never reached[12] due to a number of factors, including inaccurate or out-of-date mobile numbers and the phone being switched off or out of network. A 2015 study[13] found that one-third of women's profiles in the government registry in Rajasthan and two-thirds in Uttar Pradesh were incomplete. Data delays and late care-seeking impede overall exposure

to Kilkari. Kilkari exposure can begin in the second trimester of pregnancy, however, only 31% of subscribers answered their first call during this period.[12] Over half (58%) answered their first call after they give birth, which means they missed at least one-third of Kilkari's content.[12] Research from Bihar documented an average delay of 72 days between service delivery and being registered in MCTS.[14]

In order to understand why Kilkari was not reaching every pregnant woman as soon as possible (eg, in the second trimester), we conducted a qualitative examination of frontline barriers to the accurate, complete and timely capture of mobile phone numbers and other data in MCTS/RCH. We explored beneficiary perceptions of providing mobile phone numbers to healthcare providers, and frontline health worker (FLHW) experiences with the digital health record system broadly and capturing mobile numbers specifically. Study findings will inform health systems in India and beyond as they move towards ICT-enabled strategies to bolster the quality of HIS.

## METHODS
### Study setting
This qualitative study took place in Madhya Pradesh and Rajasthan, two large Hindi-speaking states in central and western India, respectively. These states show steadily improving but still high burdens of maternal and child mortality, significant gender gaps around technology access and literacy, and suboptimal maternal healthcare services (table 1).

Both Madhya Pradesh and Rajasthan moved to digital health records in the late 2000s, however, there are important differences between the states, in terms of the digital programmes and processes implemented (table 2). In 2016, Madhya Pradesh transitioned from MCTS to RCH. RCH added urban coverage, an initial registration of all 'eligible couples' (married couples of reproductive age) who would then be linked to pregnancy tracking when a pregnancy occurred, and the creation of village profiles.[15] Furthermore, RCH expanded the data elements collected from 111 (in MCTS) to 247 to include abortion tracking, beneficiary bank account and identification details (including the Aadhaar national identification number), additional details about the pregnant woman's antenatal care, infant feeding practice and the child's immunisation records for his or her first 5 years.[15] Mobile phone numbers were collected in MCTS and continued to be in RCH. In contrast to Madhya Pradesh, the Rajasthan state government did not adopt RCH, but instead retained its state-level version of MCTS, called the Pregnancy, Child Tracking and Health Services Management System (PCTS), which syncs with MCTS. Online supplemental annexure 1 contains an explanation of key acronyms and terms.

The states thus enable us to examine data systems in a more typical case (Madhya Pradesh), which, like most

**Table 1** Social and health indicators, Rajasthan and Madhya Pradesh

| Indicator | Rajasthan | Madhya Pradesh |
|---|---|---|
| Population | 77 million | 82 million |
| Maternal mortality ratio (deaths per 100 000)[27] | 164 | 173 |
| Under five mortality (deaths per 1000 live births)[28 29] | 51 | 65 |
| Literacy[28 29] | | |
| Female | 57% | 59% |
| Male | 85% | 82% |
| Mobile phone access[28 29] | | |
| Household ownership | 94% | 84% |
| Female access | 41% | 29% |
| Maternal health care[28 29] | | |
| Pregnant women attended antenatal care in first trimester | 63% | 53% |
| Received recommended four antenatal care visits | 39% | 36% |
| Gave birth in a health facility | 84% | 81% |
| Received postnatal health check within 2 days of birth | 64% | 55% |
| Registered pregnancies for which the mother received MCP card | 92% | 92% |

MCP, Mother and Child Protection.

large Indian states, recently moved from MCTS to RCH, and an outlier case (Rajasthan), which has retained a tailor-made electronic record system since the beginning of digitisation.

## Data collection

Five experienced qualitative researchers with master's level social science degrees (authors OU (male) MS, DG, BM and NC (female)) conducted in-depth interviews (n=59), focus groups (n=12) and observation of data collection and entry (n=9) in Rajasthan and Madhya Pradesh in September and October 2018 (table 3). All researchers were trained over a 1-week period, which included pilot testing the FLHW interview guide. In each state, we selected one district and two blocks within that district with high levels of female phone ownership to explore barriers to capturing women's mobile phone numbers in MCTS/RCH. We sampled government health system actors who were involved in MCTS/RCH at the state, district, block and frontline levels, including medical officers, data entry officers and frontline providers as well as with women who recently interacted with government healthcare providers in situations where they were asked to register their mobile numbers in MCTS/RCH (table 3).

The respondents were approached through their government health facility. A research team member contacted potential respondents by phone or face-to-face and explained the study, and that the team was from a Delhi-based company and had governmental approval, then invited them for a face-to-face meeting to learn more and, if they agreed, to participate. The study information and informed consent were read to each potential participant and then summarised in conversational language to ensure comprehension. While all the health facility staff approached for the study agreed to participate, three women invited to attend focus groups declined, sitting responsibilities at home. The interviews took about an hour and the focus group discussions (FGDs) took about an hour and a half; all were conducted in health faculty compounds and were audio recorded, and detailed notes were taken. When curious onlookers came over during focus groups and interviews, another researcher politely asked them to move on. If any supervisors, patients or family members stopped by to speak to the respondents during the interviews or focus groups the research paused until privacy was restored.

FGDs ranged from 4 to 10 participants (mean 7.6 participants). The focus group compositions broadly reflected local demographics. They included women with a wide range of education levels (from no education to master's degrees), castes (most included a mix of women from marginalised schedule caste and schedule tribal groups as well as women from 'other backwards castes' and general caste groups) and religions (three included some Hindu and some Muslim women, while the remainder were all Hindu). Most women were homemakers, while a sizeable minority worked as agricultural farmers and labourers, and also included students, tailors, shopkeepers and bangle/jewellery saleswomen.

Interviews and FGDs were conducted using semistructured guides that explored a range of domains around sharing, documenting, inputting and using data, with a focus on mobile numbers (see online supplemental file 1, eg, of the guides). We explored potential drivers of inaccuracies and delays by asking about late pregnancy identification, FLHW work environment and the relationship

**Table 2** Comparing Rajasthan and Madhya Pradesh's digital HIS

| Parameter | Rajasthan | Madhya Pradesh |
|---|---|---|
| HIS system currently in use | PCTS, a state-specific system that syncs with MCTS | RCH, an expanded and upgraded version of MCTS |
| Timeline | Adopted PCTS in 2008 (first state in India to launch an electronic health records system) and has not changed to RCH | Adopted MCTS in 2009 and changed to RCH in 2016 |
| Who collects HIS data at the frontline? | Three frontline workers: community health worker called the ASHA (Sahyogini in Rajasthan), community nutrition and preschool worker called anganwadi and ANM | Two frontline workers: ASHA and ANM. The anganwadi worker is not involved. |
| Feedback mechanisms built into the digital record system | Creates a workplan for the ANM telling her who is due for antenatal care, delivery, postnatal care as well as listing drop outs; sends SMS alerts to beneficiaries (programme called Swasthya Sandesh Sewa) | Creates a workplan for the ANM telling her who is due for antenatal care, delivery, and postnatal care, as well as listing drop outs; sends SMS alerts to beneficiaries and also to health functionaries at different levels |
| Paper forms involved in data collection for pregnancies | 1. *Pregnant women's antenatal care registry form*: Paper where up to five new pregnancies being managed by the frontline worker can be listed with details of first antnatal care visit. This form is filled in by the ASHA/ANM and passed to the data entry operator for entry into PCTS.<br>2. *Mamta card*: Card filled by ANM/ASHA; stays with the pregnant woman<br>3. *RCH register*: Book kept with the ANM or at the health centre; all pregnancies under the ANM are in this book. It is passed to the data entry operator for data entry.<br>4. *Personal ANM diary*: Unofficial paper book that ANMs fill (because RCH register is sometimes too big to bring to the village or is with the data entry operator)<br>5. *Personal ASHA diary*: Official paper book that the ASHA fills | 1. *RCH form*: One form per woman, filled by ASHA/ANM and given to the data entry operator. Meant to travel back and forth between ANM and data entry operator as the woman proceeds through her antenatal care visits and delivery.<br>2. *MCP card*: Card filled by ANM/ASHA; stays with the pregnant woman<br>3. *RCH register:* Book kept with the ANM or at the health centre; all pregnancies under the ANM are in this book. It is passed to the data entry operator for data entry.<br>4. *Personal ANM diary*: Unofficial paper book that ANMs fill (because RCH register is sometimes too big to bring to the village or is with the data entry operator)<br>5. *Personal ASHA diary*: Official paper book that the ASHA fills |

ANM, auxiliary nurse midwife; ASHA, Accredited Social Health Activist; HIS, health information system; MCP, Mother and Child Protection; MCTS, Maternal and Child Health Tracking System; PCTS, Pregnancy Child Tracking and Health Services Management System; RCH, Reproductive and Child Health.

between beneficiaries and FLHWs. In the interviews with health system actors, we also explored each step of a detailed description of data flow (figure 1) from when a pregnant woman first interacts with an FLHW until her health information is entered into the online portal and beyond to understand perceptions on the use of this data.

### Data analysis

Daily debriefs enabled the team to share emergent findings, refine the focus of their probing for the next day's data collection and identify areas of saturation. The audio files were transcribed and translated into English. Data were coded in Dedoose by OU and KS, using principles of thematic network analysis.[16] A coding framework was developed that consisted of emergent codes on specific reasons for inaccurate and delayed data, which were then grouped according to an overarching data flow framework (figure 1). For instance, we created a code cluster for late antenatal care registration, which included codes to be applied to text describing when and how pregnancies

come to the FLHW's attention, when pregnancies were entered into the online portal, reasons FLHWs become aware of pregnancies late (after the first trimester) and the implications of a woman's choice of the public or private sector for antenatal care on timeliness of registration. After coding, we read the text excerpts that had been tagged for specific codes to identify how key actors experience of the data collection and digitisation process, reasons for late or inaccurate data and mechanisms that can bolster timeliness and accuracy.

### Conceptual framework

While we initially set out to understand inaccuracies and delays in entering mobile phone numbers into the pregnancy registry, it became clear that this one piece of data could not be separated from the broader data collection process at the frontlines of government health service provision. We, thus, examined mobile phone number data collection and digitisation within the context of an overarching health system data flow framework (figure 1).

**Table 3**  Respondent sample

| Respondent type | Respondent profile | MP | RJN | N |
|---|---|---|---|---|
| IDIs | | Number of IDIs | | |
| State level stakeholders | Senior government employees of the department of health and family welfare who are in charge of data | 3 | 1 | 4 |
| District stakeholders | Community mobilisers, District Programme Manager, District Community Mobiliser, District Monitoring and Evaluation Officer, District Nodal Officer | 3 | 4 | 7 |
| MOs at primary health centres (PHCs) | Doctors (allopathic, homeopathic or ayurvedic), 5.5 years training | 2 | 4 | 6 |
| DEOs | Information technologists with undergraduate level education | 4 | 6 | 10 |
| ANMs | Female maternal and child health worker with 1.5 years training (6 months midwifery focused) | 10 | 9 | 19 |
| ASHA community health worker | Female volunteer community health worker, received incentive-based remuneration, initial 24 days training and periodic week-long additional training | 3 | 5 | 8 |
| Other stakeholders at the PHC and block level | Block Programme Manager, Multipurpose Health Supervisor, Primary Health Centre Supervisor | 2 | 3 | 5 |
| Total IDIs | | 27 | 32 | 59 |
| FGDs | | Number of FGDs | | |
| Beneficiaries | Pregnant and post-partum women with mobile phones who recently interacted with health system actors for MCTS/RCH registration | 6 | 6 | 12 |
| Observation | | Number of observations | | |
| Beneficiary–FLHW interaction | Observation of beneficiary–FLHW interaction (antenatal or post-partum) where MCTS/RCH data recorded | 1 | 1 | 2 |
| Data entry into electronic system | Observation of data entry into electronic system | 3 | 4 | 7 |
| Total observations | | 4 | 5 | 9 |

ANM, auxiliary nurse midwife; ASHA, Accredited Social Health Activist; DEOs, data entry operators; FGDs, Focus Group Discussions; FLHW, frontline health worker; IDIs, in depth interviews; MCTS, Maternal and Child Health Tracking System; MOs, medical officers; PHC, Primary Health Centre; RCH, Reproductive and Child Health.

This framework identifies six components that enable the creation and movement of health data, including mobile phone numbers, from beneficiaries through FLHWs to electronic data entry and onward.

The first component, 'beneficiary', describes women's access to the data required for documentation (e.g., whether they have a mobile phone number to provide) as well as their attitudes towards sharing this data with government health system functionaries (e.g., their willingness to provide their mobile numbers when asked). The second component considers the beneficiary-FLHW interactions that initiate the government health system's awareness of a health event. In this section, we consider how and when FLHWs become aware of a new pregnancy, the value that FLHWs place on collecting beneficiary data and the strategies used to collect data. The third component involves the creation of initial paper records. In this section, we consider the various official and unofficial

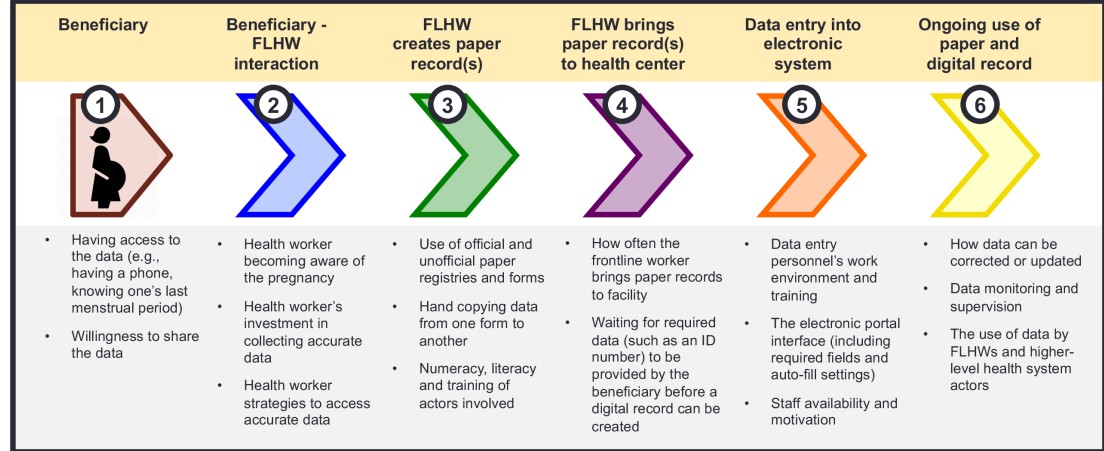

**Figure 1**  Data flow framework for electronic health record systems. FLHW, frontline health worker.

forms and registers where data are recorded, the health worker's literacy and numeracy if data are copied from paper form to paper form, and potential delays between a health worker interacting with a beneficiary and a paper record being created. The fourth component considers the process through which a paper health record reaches the site of digitisation. We consider how often the FLHW visits the digitisation site and whether the FLWH waits to bring paper records for digitisation until all mandatory fields have been filled out. While steps 3 and 4 may be dropped as FLHWs directly create digital records themselves, in many health systems in lower resource settings, paper records are still created during outreach service provision. The fifth component is the time when data are digitised and considers the data entry personnel's work environment, training, the electronic portal interface and staffing considerations. The final component examines the ongoing use of data through paper forms and online systems. Here, we consider how data can be corrected or updated, data monitoring and supervision and the use of data by FLHWs and higher level health system actors.

### Patient and public involvement

The research was shaped by health system actor priorities, experiences and preferences through iterative probing and flexibility within our research domains. Results were disseminated to Government of India stakeholders but not to research participants due to the policy-level implications of our findings.

### FINDINGS

Our findings are organised conceptually according to the six components in the general data flow framework. For each step, we provide a description of how this aspect of the data system is experienced by the actors involved and identify barriers and facilitators to timeliness and accuracy.

### Level 1. Beneficiary

#### Women faced barriers to accessing phones and services that generate the data required by MCTS/RCH

Women who did not have their own personal mobile phones could not provide their number. Many women lived in households with phones, so could provide their husband's phone number or the number for a shared phone. Women living in households without a phone could provide a neighbour's phone number or a FLHW's number.

> R: Some women have their husband's mobile number. In some cases both husband and wife don't have mobile phones. Some are from poor families where both of them don't have mobile phones.
>
> I : So, what do you do in such situation?
>
> R: Either we take a relative's phone number or a neighbor's. […] If any laborer is there who does labor work how will he have mobile phone? Even if it's

there then her husband must have taken it while going for labor work. (MP_IDI_FLHW_06, ANM)

Additional data required from beneficiaries included their bank account numbers and branch names (for both the beneficiary and her husband), national identification (Aadhaar), ration card number and state identification numbers (Bhamashah state health insurance in Rajasthan and Samagra in Madhya Pradesh). FLHWs explained that women often did not have all of these items or did not know this information. Accredited Social Health Activist (ASHAs) and Auxiliary Nurse Midwives (ANMs) were tasked with following up with women to encourage them—or at times personally help them—open bank accounts, register for identification numbers and provide photocopies of this information.

> Like the [bank] account number, Aadhaar number we get them very rarely. We have to with the ASHA then go again to get information. We fill the form and submit it and it's fed in. If half the information remains incomplete then we can't do the RCH. (MP_IDI_FLHW_01, ANM)

#### Women were generally comfortable sharing data with frontline health workers

Women and FLHWs overwhelmingly reported that beneficiaries with access to a mobile phone shared their mobile number with their ASHA or ANM without hesitation and without questioning. They explained that women trusted FLHWs and felt that the number would only be used to help them, such as to convey important information about health services or financial benefits through government programmes.

> They are observing me from 20 years so they are aware that madam will not misuse the number so they never deny giving number. (RJN_IDI_FLHW_25, ANM)
>
> [Frontline health workers take our mobile number] so that they can give us a phone call next time to call us whenever vaccination is to be given. That's why. (RJN_FGD_BENEF_07)
>
> They are aware that the mobile number is essential as when the money will come in the bank account then they will get the message on mobile. We take the mobile number for their comfort only as we can call them as required. […] And calls also come on their mobile who tells them about the health. (RJN_IDI_FLHW_27, ASHA)

There were exceptions to this predominantly positive view of sharing mobile numbers with health workers. FLHWs and beneficiaries reported that a very small minority of women resisted sharing their numbers with them because of concern about receiving calls from strangers or because they had hidden the fact that they had a mobile phone from other family members.

In Madhya Pradesh, most women thought that only the ASHA had their number. When told that their numbers were

passed onwards from the ASHA to the ANM, women were unconcerned. In Rajasthan, women were generally aware that their mobile numbers were collected for use beyond beneficiary-ASHA communication. They clearly explained that their mobile number was linked to their receipt of financial benefits. While most women in Rajasthan said the only calls they received from government actors were from their ASHA, calling them for checkups, a few received informational calls from the government of India, which were likely Kilkari messages, and one said that the government would use their mobile numbers to call them and check 'whether the facilities which government is providing are reaching us or not' (RJN_FGD_BENEF_09).

### Level 2. FLHW-beneficiary interaction
#### Early identification of most pregnancies—with notable exceptions
Digital records of pregnancies can only be created after the government health system functionaries identify a pregnant woman. ASHAs are a pregnant women's first point of contact with the government health system. ANMs rely on ASHAs to identify pregnant women and encourage them to meet the ANM for antenatal care. Beneficiaries and FLHWs reported that ASHAs were notified within the first 4 months about almost all pregnancies.

However, FLHWs noted that women living in remote communities, migrants and less educated women, may delay seeking antenatal care due to 'carelessness' (RJN_IDI_FLHW_27, ASHA), superstition and not knowing that they were pregnant. Wealthier women who received antenatal care in private facilities would also be missed by the government health system or registered only after giving birth.

#### FLHWs place high value on collecting and digitising beneficiary data
FLHWs in Rajasthan and Madhya Pradesh articulated a clear focus on accurately collecting and digitising women's bank account details and identification numbers, since this data enabled women to receive financial support from government programmes. While phone numbers were considered important for ASHA-beneficiary communication, the value of digitising this information (rather than just ensuring it was collected in the ASHA's paper records) was not always clear. Some health workers in Madhya Pradesh could only articulate a vague sense that entering mobile numbers in RCH was useful. Most FLHWs in Madhya Pradesh reported that mobile numbers were used by the state or central government to call beneficiaries and check whether they are receiving appropriate health services, suggesting that FLHWs saw the digitisation of women's mobile numbers as a means for the government to keep tabs on FLHW performance. In Rajasthan, FLHWs were clear about the value of mobile numbers in PCTS and explained that the numbers were not only used for checking that women were receiving health services but also for conveying health information, advertising government programmes and notifying women about financial transfers.

#### FLHWs describe several strategies for trying to collect accurate data
In Rajasthan and Madhya Pradesh, ASHAs and ANMs explained that many women could not accurately recall their mobile numbers at antenatal care visits. For cases when women did not have mobile phones with them, ASHAs would visit their homes and ask other family members to provide a mobile number or ASHAs and ANMs would ask women to bring a mobile number written on a piece of paper to the next antenatal care check-up.

> Like, they don't know, some of them… they don't know the mobile number. So, we tell them to ask the people in her house who can write, to note it down on a piece of paper, and bring it to us. Tell them the sister (nurse) needs it, for your information. So they do that (MP_IDI_FLHW_01, ANM).

ASHAs were confident that they received accurate numbers because they reported frequently calling women using the number provided. In cases when women had mobile phones with them, some ANMs or ASHAs reported giving themselves a missed call from beneficiary mobile phones to identify the number; however, the use of missed calls was not a universal practice, as observed during beneficiary—FLHW interactions.

FLHWs said that pregnant women sometimes asked FLHWs to record their husband's mobile number rather than their own, even when she had her own phone. Women in one FGD (RJN_FGD_BENEF_10) explained that the husband's mobile number was often provided, so that he would receive the Short Message Service (SMS) notification of a financial transfer into the beneficiary's bank account. While some FLHWs complied in taking the husband's number, others tried to convince each woman to provide her own mobile number since her husband will frequently be out of the house with the mobile.

FLHWs reported asking women to provide photocopies of bank account information and government identification cards, in order to reduce errors that could be introduced by women conveying this information on scraps of paper. When assessing last menstrual period in order to estimate gestational age, two FLHWs noted that many women struggled to report exact dates. FLHWs estimated last menstrual period based on women's recall of the moon cycle, harvest or festivals, which would introduce inaccuracy around pregnancy stage.

### Level 3. Creation of paper records
#### Beneficiary data copied by ASHAs and ANMs across multiple paper forms and registers
In Madhya Pradesh and Rajasthan, when an ASHA identified a pregnancy in the village, she would write key information about the woman in her official ASHA diary. In cases where ASHAs were considered sufficiently literate by the local ANM, the ASHA was also expected to start a paper RCH form (in Madhya Pradesh) or ANC form (in Rajasthan) by filling in some initial fields. In Madhya Pradesh, we found that ASHAs in one block reported

being paid Rs. 50 (US$0.80) for each form they created, while in a nearby block ASHAs received no payment. In both states, some ASHAs first wrote the pregnant woman's details in an unofficial notebook and then copied this information into the official ASHA diary and the RCH/ANC form. While some ASHAs were clear and confident about their role filling in the RCH/ANC form, others were unsure about which documents they completed.

When the woman comes the village health and nutrition day (an outreach event where antenatal care is provided by ANMs), the ASHA will pass the partially filled RCH/ANC form to the ANM. The ANM will fill in additional details from the woman's first antenatal care visit, often with the ASHA's help. ANMs explained that they were overwhelmed by documentation requirements and, thus, had to take assistance from ASHAs, even when ASHAs struggled with the literacy and numeracy requirements of this work.

> We have to fill a big register, we make sehyogini (ASHA) fill it; that is the reason it gets wrong. Now they (ASHAs) don't understand. They themselves have filled the card. Now, there was so much crowd that time, that I was writing something… and if they say, the weight is 53 and she says its 61, so then there's confusion like that. So sometimes they write it wrong. When there's a crowd sometimes, we take help from them only, and it becomes wrong. (MP_IDI_FLHW_07, ANM)

In addition to the RCH/ANC form, ANMs record data about pregnant women in the ANM RCH register (a book) and create a Mother and Child Protection (MCP) card (Madhya Pradesh) or Mamta card (Rajasthan). MCP/Mamta card is given to the pregnant woman to keep. Similar to our observations of ASHAs, ANMs in Madhya Pradesh also sometimes wrote down beneficiary information in unofficial notebooks and then copied this information into their official RCH register and the RCH forms once they returned home; they explained that the RCH register is heavy and they wanted to keep it neat and clean. In Rajasthan, the official register was smaller and more portable, so ANMs tended to carry it with them. In both states, we heard of instances when official registers were kept in health facilities for review or digitization, which further necessitated the use of unofficial notebooks.

### Delays in antenatal care have implications for the creation of health records

There could be several weeks or even months between the ASHA becoming aware of a pregnancy and the ANM meeting the woman for first antenatal care. No paper record could be completed until the woman interacted with an ANM for antenatal care, thus introducing delay in the production of official health records.

### FLHWs found data collection under RCH to be too time intensive

ANMs in Madhya Pradesh said that they found data collection to be highly burdensome. They felt that their work

had shifted from healthcare provision to finding documents and filling in forms. FLHWs described facing pressure to complete the digital records from senior actors in the health system hierarchy as well as from beneficiaries who were worried about delays in receiving their financial incentives. They described having to fill up to 70 data fields in various paper forms and registers, and having to track down beneficiary banking and identification details, without adequate time or support.

> We have to do this also and that also. In this what happens is, like this "Prashuti Sahayata Yojana" has come [i.e., yet another scheme has been introduced]. So, we keep on calling them the whole day that "Get this paper" "This is not enough" "This paper is short." Just now also madam [a supervisor] called. And sir [another supervisor] also called saying that high risk pregnancy too needs to be checked too. (MP_IDI_FLHW_06, ANM)

> They've increased our work a lot, a lot. … We need to visit each beneficiary 10 times… "Give the Aadhar card, the Samagrah ID." The beneficiary isn't able to submit, isn't able to give it to us, and above that they reach here and start complaining, that, madam isn't filling our form so we aren't getting our money when they themselves aren't giving us the documents all together. We used to go to each house, of the beneficiary, give them the vaccinations, meet each one. And the bai [women] that were there, they had very good behaviour, were respectful, and would make us sit inside, that come, madam, sit inside. Would treat us very well. And, these days I don't know what is happening, so the beneficiaries also snap at us, for a little bit. If they say, madam we need the money, and we tell them to submit the documents, and they'll get the money… Oh oh, but you need this paper, and that paper, and we've given it to the Anganwadi, to you, and to the ASHA, what is this nonsense. (MP_IDI_FLHW_07, ANM)

ANMs who worked with low-literacy ASHAs noted that they faced an additional struggle because they could not rely on the ASHAs to complete any paper forms.

### Level 4. Paper records brought to health centre
Delay between creation of paper records and records reaching the health centre with a computer

The ANM becomes aware of a pregnancy during the woman's first antenatal care visit and considers the woman's pregnancy to have been registered at this time. However, the pregnancy is not entered into the online portal until the ANM brings the RCH/ANC forms and RCH registers to the health centre for digitisation. This occurs only once per month. In addition, if an ANM is waiting for a beneficiary to provide documents required to complete the RCH/ANC form, she will hold on the form even longer, until a future visit to the health centre.

There are some women in the village, even after their delivery, their Samagra ID isn't able to be made…[…] [So] It's not fed [entered into RCH] here because the Samagra ID is incomplete (MP_IDI_FLHW_01, ANM).

## Level 5. Data entry
### DEOs spoke of high workloads and some technical challenges but noted recent improvements and supportive peers
Data entry operators (DEOs) in both states said there was excess workload, which could lead to data entry backlog. They explained that in addition to entering maternal and child health data, they were also responsible for entering data from the pharmacy, ASHA records, birth and death registration, Health Management Information System (HMIS) and health facility forms. In some areas, one DEO was covering the workload of two or three because of vacancies. ANMs suggested that DEOs may make data entry errors because they were rushing or careless.

In both states, the DEOs mentioned that the infrastructure to support them was improving, such as the provision of newer computers, faster internet and upgraded servers. In Rajasthan, the DEOs had battery backup for power outages and a dedicated broadband internet connection; even so, during observation, opening the PCTS portal took up to 10 min because of slow internet connection or server capacity. In Madhya Pradesh, many DEOs did not have functioning battery backup, leaving them unable to work when the electricity went off. Some relied on their own portable WIFI device or their mobile phone's hotspot, both of which were often slow.

DEOs in Madhya Pradesh reported having received only one training on RCH, while DEOs in Rajasthan received frequent trainings and felt reasonably well supported by the supervisory structure as well as their peers. The one-time training in Madhya Pradesh occurred years ago, and many months before RCH was introduced, so recipients forgot much of it. It consisted of a presentation without any hands-on learning. DEOs reported learning how to use RCH on the job, through trial and error, and from other DEOs, who provided ongoing peer support.

### Required fields and auto-fill invite errors
The digital portals had numerous required fields, validation checks and autofill features, including that 10 digits be entered in the mobile number field. In many cases, these features supported accurate and complete data entry. However, DEOs recalled instances where these features forced them to introduce errors into the system. For instance, when DEOs encountered forms that did not have a beneficiary mobile number, DEOs reported using the ASHA's number or their own number because the mobile number field is mandatory. In Madhya Pradesh, DEOs reported that RCH initially required all mobile numbers to start with the digit 9. When a new phone company began issuing mobile numbers beginning with 6 or 7, DEOs could not get the system to accept these numbers until the RCH portal was updated.

In RCH, pregnancy records are linked to the eligible couple record. The pre-existing eligible couple registry auto-completes fields in the new pregnancy registry, including for the mobile number. Thus the phone number provided by a family at time of marriage becomes the default current phone number at the time of pregnancy.

### DEO and ANM absences can lead to data entry delays and may compromise quality
When DEOs took time off, there were not always additional trained personnel to fill in. Other health system actors (such as an ASHA supervisor) would take over data entry, even without training. In cases of longer term absence, such as when there was no DEO appointed to a PHC at all, a DEO at a neighbouring facility would be assigned to cover both catchment areas. ANMs then had to bring their paper records to a more distant health facility to have them digitised and DEOs had to cover double the caseload.

There are gaps in data collection at the frontlines when ANMs take time off, during transition from an older ANM to a new recruit, or when the ANM position is vacant. During these periods, no health records are created or digitised, but the district health system actors demand that data continue to be entered. This leaves the DEO in a difficult situation, wherein no paper records are being passed on to them, but they are expected to create digital records.

## Level 6. Ongoing use of paper and online records
### Mobile numbers are not corrected or updated in the online system
Beneficiaries explained that they updated their ASHA whenever they changed mobile numbers, and ASHAs were confident that they retained up-to-date mobile numbers for the pregnant women in their villages. However, ASHAs, ANMs and DEOs reported that they do not update or change mobile numbers in the online portals. Some DEOs were unsure if it was even possible to do so. Checking and correcting bank account details and government identification details was common, since this ensured financial transfers proceeded. Ensuring correct and up-to-date mobile numbers was not a priority. Even among women and FLHWs who identified benefits for women who shared their mobile number with the government, the value of receiving government information and notifications was overshadowed by the importance of receiving financial incentives.

### Supervision focuses on completeness not accuracy
DEOs as well as block, district and state level actors focused on data entry completion and timeliness but not on accuracy. DEOs were encouraged to complete their data entry activities each month and clear backlog. However, attention to data accuracy only arose when mistakes came to light through irregular means, such as a financial transfer going into the wrong bank account or a polio outbreak alert being triggered by a DEO incorrectly entering eight polio cases rather than zero.

In Madhya Pradesh, some health facilities had multipurpose supervisors and ASHA support staff who checked the forms and registers provided by ANMs and ASHAs to the DEO and who oversaw the DEO's work. These supervisors could flag potential inaccuracies for follow-up. However, in other facilities, these positions were vacant. The medical officer was expected to sign off on the DEO's work but was so busy with patient care that they often gave an automatic approval or even provided their login and password to the DEO, so that the DEO could approve their own work.

In both Madhya Pradesh and Rajasthan, DEO work at the health centres was overseen by block-level supervisors. In both states, these actors reported being overwhelmed by having to oversee five to seven software programmes and more than 50 reporting processes and could only focus on issues of incomplete data.

At the frontlines, ANMs and ASHAs in Madhya Pradesh and Rajasthan had no way of knowing whether the mobile number entered in PCTS or RCH was correct and had never been asked to cross check these digital records with beneficiary numbers.

### Health system actors have an overall positive view of the value of digitisation

Despite noting hiccups and increased documentation burden, ANMs and ASHAs were generally positive about the move towards digital health records. They framed the use of digital systems as a better way for them to showcase their hard work and to receive recognition (and, for ASHAs, performance-based remuneration), rather than a way to benefit from the digital system's ability to generate ANM workplans or other synthesised data. Block, district and state-level actors described benefiting enormously from the digital system. The explained that it enabled them to see the progress of programmes, monitor service provision by FLHWs, conduct yearly planning to address gaps and prioritise resources: 'In one click the full report can be retrieved' (RJN_IDI_MO_04, doctor).

### DISCUSSION

This study explored how beneficiaries, frontline health workers, DEOs and higher level actors in the government health system experienced HIS digitisation to understand barriers to accurate, complete and timely capture of mobile phone numbers and other data (table 4). We found that frontline workers and their supervisors prioritised the accuracy of administrative data elements that enabled pregnant clients to receive government financial entitlements (e.g., bank account information, identification numbers) over mobile phone numbers. Providers reported that it often took months—and significant effort that detracted from healthcare provision – to gather the documents and information from pregnant clients required to create a digital record; frontline workers relied on their existing paper systems to provide ongoing

healthcare and keep track of up-to-date beneficiary mobile numbers.

Scholars have critiqued India's health system digitisation as disproportionately focused on the collection of data rather than its use.[17 18] However, the health system actors that we interviewed conveyed a clear understanding of ways in which the digital HIS supports work planning and monitoring. Nonetheless, the overarching logic behind digitisation articulated by frontline workers remained as a mechanism to feed information up the health system hierarchy rather than to improve their work planning or to access information about their clients.

While pregnant women were willing to share their or their family's mobile numbers with ASHAs and ANMs, many were unaware that their mobile numbers were passed onward and entered into computers. Although women trusted the health workers and associated the provision of requested data with receiving benefits from the government, strengthening consent processes can protect beneficiaries and safeguard relationships at the frontlines. These processes can inform beneficiaries and frontline providers about the benefits of ensuring accurate mobile numbers are registered in the digital system (including that beneficiaries can be subscribed to mHealth programmes such as Kilkari), potentially increasing both parties' motivation to keep the registry up-to-date.

As identified earlier by other researchers,[13] we found poor standardisation across multiple data systems (paper based and digital) and identified opportunities for streamlining records and entry options to become more intuitive and user friendly. Echoing findings from other settings,[19] supervision focused on completeness rather than accuracy, highlighted by the fact that there were no mechanisms to check whether mobile numbers were correct. Workers at the frontlines require clear guidelines on how to handle common challenges in accessing data, particularly mobile phone numbers, including how to handle data changes (such as new phone numbers) and unavailable data (such as no phone number). Health workers require training on the value of maintaining an up-to-date mobile number in their data registries, so that they are motivated to ensure that these numbers are correct. Furthermore, in light of research findings that 'SIM churn' (changing SIM cards, and as a result getting a new mobile number) is as high as 44% per year among rural families,[10] frontline workers need to regularly confirm beneficiary mobile numbers and have access to user-friendly mechanisms to update numbers whenever they change.

Required fields, auto-fill features and validation checks can force DEOs to adjust the data provided to them according to their judgement in order to proceed with data entry. Another study on PCTS in Rajasthan found that the portal would not accept delivery dates that appeared to occur after more than 9 months of gestation, a situation created by incorrect gestational age estimation based on faulty last menstrual period estimates.[20] Finally,

**Table 4** Summary of barriers to the creation of timely, accurate and complete mobile phone records in MCTS/RCH data

| | Barriers to timely data | Barriers to accurate and complete data |
|---|---|---|
| 1. Beneficiary | ► Beneficiaries must open bank accounts and attain government identification before an electronic record can be created. | ► Duplicate mobile phone entries are created when one number is provided by multiple pregnant women, such as when multiple women in a joint family with one phone become pregnant, or when multiple women provide a neighbour's number or the ASHA's number.<br>► Non-beneficiary numbers are entered when women without personal mobile phones provide their husband's number, a shared family phone number, a neighbour's number, or the ASHA's number. |
| 2. Beneficiary–FLHW | ► While ASHAs learn of most pregnancies within the first trimester, pregnancies among migrants, women living in remote communities, very poor women who did not see the value in seeking early antenatal care, and wealthy women who received antenatal care in the private sector were all detected late—sometimes even after the baby was born. | ► FLHWs value collecting correct mobile numbers so that ASHAs can keep in touch with beneficiaries. The value of accurately digitising these numbers is not always clear to FLHWs in Madhya Pradesh. In both Rajasthan and Madhya Pradesh, accurate mobile phone number digitisation does not have immediate or direct influence on healthcare or financial transfers.<br>► While ASHAs and ANMs have a number of strategies to seek correct mobile numbers, beneficiaries may incorrectly recall their mobile numbers; checking the beneficiary's number through a missed call is common but not universal and is not an official requirement.<br>► Accurate gestational age estimation (which is linked to the provision of stage-based information in Kilkari) is a challenge since women cannot always report the exact date of their last menstrual period. |
| 3. FLHW creates paper records | ► There can be a delay of weeks or even months between an ASHA learning of a new pregnancy and the woman receiving her first antenatal care from the ANM at village health and nutrition days.<br>► It is only after the woman meets the ANM that her (paper) antenatal care/RCH form can be filled in with details of the first antenatal care visit. | ► Copying errors can occur when ASHAs and ANMs write women's information in multiple places for different programmes, often with slightly different data fields; beneficiaries may also provide their mobile phone numbers on scraps of paper.<br>► ASHAs are often expected to complete many fields in the RCH/ANC form before the ANM fills in additional details and passes it to the DEO; ASHAs with lower literacy struggled with this responsibility.<br>► FLWHs in Madhya Pradesh found data collection for RCH to be highly burdensome and described high pressure from above to fill forms and registries without adequate time or support.<br>► Some ASHAs were confused about the names and purposes of various paper forms and some hand-made their own additional unofficial registries that they used in the field. |
| 4. FLHW brings paper records to health centre | ► Many ANMs bring paper antenatal care/RCH forms and registers to the health centre for data entry only once a month.<br>► A woman's paper form cannot be passed to the health centre until she has provided all required fields (such as a bank account number), which may introduce further delays. | ► Respondents did not note any risks to data accuracy or completeness while transporting paper forms and registers from the field to the data digitisation facility. |
| 5. Data entry into online portal | ► Server errors, internet and electricity issues may delay DEO in creating electronic records.<br>► If DEOs are overburdened with data entry activities, they can fall behind. | ► When entering a pregnancy into the online portal, RCH auto-fills the mobile number provided at the time of 'eligible couple' registration; this number may be outdated.<br>► DEOs in Madhya Pradesh found their training on RCH to be insufficient; peer support enabled DEOs to navigate challenges. DEOs in Rajasthan were confident in PCTS and received frequent training.<br>► When DEOs are on leave or when DEO positions are vacant result in other health facility employees, who have not been trained, perform data entry. |
| 6. Ongoing use of paper and online records | ► None identified | ► Although the ASHA is generally notified if a pregnant woman's contact number changes over the course of her pregnancy or postpartum, this new number will not be updated in the PCTS/RCH portals.<br>► Supervisor positions remain vacant in some health facilities and higher level supervisors (block and district level) oversee numerous databases and reporting systems.<br>► Supervisors tend to focus on completeness and timeliness rather than accuracy. |

ANM, auxiliary nurse midwife; ASHA, Accredited Social Health Activist; DEOs, data entry operators; FLHW, frontline health worker; MCTS, Maternal and Child Health Tracking System; PCTS, Pregnancy Child Tracking and Health Services Management System; RCH, Reproductive and Child Health.

the consolidation and standardisation of multiple records may reduce errors introduced by manual copying from form to form and reduce the work burden on frontline actors.

Three core determinants of accurate, complete and timely administrative data emerged from our research and are likely applicable to health registries for pregnant and postpartum women in other lower resource settings. First, beneficiaries who trust their health workers and the government more broadly are willing to provide accurate data.[21 22] Second, data entry systems that are easy to use, appropriate to the local context and useful to the frontline worker facilitate data accuracy and timeliness.[8 23] And third, supportive supervision and adequate resources are vital to building frontline health worker skills and knowledge across clinical and administrative functions.[24–26]

This study has some limitations, which point towards areas for future research. The information presented here was

either gathered through interviews and focus groups and, thus, self-reported, or was gathered through observation, wherein DEOs and FRHWs were aware that their work was being observed. Thus, respondents may have minimised or hidden unauthorised behaviour, such as shortcuts to speed up data entry or unsanctioned task shifting. Future quantitative research could more objectively measure data delays (e.g., time elapsed from first antenatal interaction to the creation of a digital record) and data accuracy (e.g., systematically validating key data fields, including mobile phone numbers). For this study, we spoke to pregnant and postpartum women with mobile phones who recently interacted with health system actors: this sample enabled us to understand reasons for delays and inaccuracies among beneficiaries whose data should have been easy to digitise. Future research among women without mobile phones and women who interacted with the government health system many months after childbirth would triangulate understanding about the data sharing practices of these additional populations. Furthermore, research with additional beneficiaries could provide richer insight into equity dimensions related to different caste, religion, socioeconomic status or educational levels. Such insights would best be gleaned through targeted in-depth interviews with specific marginalised populations and FGDs that separated women by caste, religion, socioeconomic status or educational levels.

## CONCLUSION

Frontline health workers are broadly supportive of the move towards digitisation despite implementation challenge. Beneficiary willingness to share their information with health workers—including their mobile phone numbers—highlights their trust in health workers and the value they place on government financial incentives. This positive engagement bodes well for derivative uses of HIS, such as for mHealth programmes and surveys. However, the underlying causes of inaccuracies and severe delays must be addressed before the true benefit of these uses will be realised. In addition to data system and supervisory supports, frontline workers and beneficiaries themselves must understand and experience the benefits of accurate, complete and timely digital HIS.

**Author affiliations**
[1]Department of International Health, International Health, Johns Hopkins Bloomberg School of Public Health, Baltimore, Maryland, USA
[2]Oxford Policy Management, New Delhi, Delhi, India
[3]BBC Media Action, Delhi, Delhi, India
[4]Independent Researcher, New Delhi, Delhi, India
[5]Centre for the Study of Law and Governance, Jawaharlal Nehru University, New Delhi, Delhi, India
[6]School of Public Health and Family Medicine, University of Cape Town, Cape Town, Western Cape, South Africa

**Acknowledgements** The authors are grateful to the health system actors who generously provided their time and insights to make this research possible. This work was made possible by the Bill and Melinda Gates Foundation. We thank Diva Dhar, Suhel Bidani, Rahul Mullick, Dr. Suneeta Krishnan, Dr. Neeta Goel and Dr. Priya Nanda for giving us this opportunity. We additionally thank Vinit Pattnaik at OPM and

Erica Crawford at Johns Hopkins for their support to the financial management of our work.

**Contributors** KS, SC, OU, DM and AEL conceptualised and designed the study. MS, DG, BM and NC conducted the data collection and preliminary data analysis through daily analytic debriefs. KS and OU led and managed the data collection and analysis, including the coding and thematic synthesis. KS drafted the manuscript and revised it based on critical and substantive input from SC, OU, AEL, MS, BM, DM and NC. All authors agree to be accountable for all aspects of the work related to accuracy and integrity. KS is the guarantor who accepts full responsibility for the work and the conduct of the study, had access to the data, and controlled the decision to publish.

**Funding** This work was supported by the Bill and Melinda Gates Foundation grant number OPP1179252.

**Competing interests** None declared.

**Patient consent for publication** Not applicable.

**Ethics approval** This study involves human participants and was approved by the institutional ethics review boards at Sigma, Delhi, India (10041/IRB/D/17-18) and JHU, Baltimore, USA (00008360). All respondents provided informed oral consent. Participants gave informed consent to participate in the study before taking part.

**Provenance and peer review** Not commissioned; externally peer reviewed.

**Data availability statement** Data are available upon reasonable request. Data for this study consist of qualitative interview and focus group discussion transcripts. Uploading all transcripts for open availability would compromise our ability to fully mask participant details. However, we are happy to share anonymised portions of these transcripts upon reasonable request.

**ORCID iDs**
Kerry Scott http://orcid.org/0000-0003-3597-9637
Osama Ummer http://orcid.org/0000-0002-4189-5328
Sara Chamberlain http://orcid.org/0000-0003-4785-6482
Namrata Choudhury http://orcid.org/0000-0001-9019-4643
Amnesty Elizabeth LeFevre http://orcid.org/0000-0001-8437-7240

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
