## [Reviewer comments · BMJ Open]

ARTICLE DETAILS

TITLE (PROVISIONAL)	At the frontlines of digitization: a qualitative study on the challenges and opportunities in maintaining accurate, complete, and timely digital health records in India's government health system
AUTHORS	Scott, Kerry; Ummer, Osama; Chamberlain, Sara; Sharma, Manjula; Gharai, Dipanwita; Mishra, Bibha; Choudhury, Namrata; Mohan, Diwakar; LeFevre, Amnesty

VERSION 1 – REVIEW

REVIEWER	Sullivan, Clair The University of Queensland Faculty of Medicine and Biomedical Sciences, Centre for Health Services Research
REVIEW RETURNED	06-May-2021

GENERAL COMMENTS	This analysis aims to identify barriers to the accurate, complete and timely creation of MCTS and RCH HIS records in India. We explore beneficiary perceptions of providing mobile phone numbers to health care providers, and frontline health worker (FLHW) experiences with the digital health record system. Study findings will inform health systems in India and beyond as they move towards ICT-enabled strategies to bolster the quality of health information system Thank you for this paper on such an important topic : and presumably even more relevant given the current surge of COVID19 in India. Overall I think this paper is interesting, however there seems to be two competing aims.
--

	1.Experiences of FLHW with digital health records in 2 Indian states 2.The issues surrounding collection of mobile numbers to enable and deliver timely maternity care I am not sure why they are admixed in this paper? It makes the paper confusing, with an emphasis on mobile phone numbers making a clear clinical information system focus difficult. We do not really explore the issues of clinical data collection in depth (mobile phone numbers are not clinical information). Then there is significant discussion around the delivery of maternity financial benefits and telehealth maternity education and advice (again not a usually function of clinical information systems). I suggest two papers: 1.exploring the experiences of clinical information systems in the 2 states 2.the experiences of delivery digital enabled maternity care in the 2 states As currently presented, it is difficult to draw coherent conclusions on either from this work. Thank you for addressing this important topic in a low resource environment. Minor comments: p14 line 13 please correct spelling error “because they ould not rely on the ASHAs to complete any paper forms” p17. This sentence seems to contradict itself: please review. “We found that frontline workers and their supervisors prioritized the accuracy of core health information
--	---

	and of data elements that enabled pregnant clients to receive government financial entitlements (e.g., bank account information, identification numbers) over timeliness or the accuracy of beneficiary mobile phone numbers” p18 there is an overwhelming focus on mobile phone numbers. I suspect perhaps I don't understand the context to understand why the mobile numbers are so important. Mobile numbers are core administrative data and are not really part of the clinical record. Mobile number collection can be part of a patient administrative system (PAS) rather than an electronic medical or health record. (EMR/ EHR)
--	--

REVIEWER	Urtaran-Laresgoiti, Maider University of Deusto, Deusto Business School Health
REVIEW RETURNED	28-Jul-2021

GENERAL COMMENTS	Dear authors, The submitted manuscript is interesting and gives a valuable new perspective of the barriers to the accurate, complete and timely creation of health information systems used for maternal, child, and reproductive health in India. Although its appropriateness for its publication, I suggest some points have to be reviewed and modify in the manuscript before its acceptance. Please, be aware that page reference in the following lines is taken from the top side of the pages. OVERALL REMARK:  - I would include a dictionary (Annex 1) with the most repeated and relevant acronyms in the manuscript to facilitate comprehension of all included terms. Please, consider also to add the complete description of all acronyms at least once in the manuscript (the first time they appear). METHODS:  - Page 6, lines 46-50: As it is stated “the states provide a natural comparative study”. In this sense, I would include this piece of information in the last paragraph of the Introduction section, where the objectives of the paper are presented. Furthermore, if the comparison is allowed and making a comparison is also an objective of the research, it would also be interesting to present data in the results section in a comparative way, highlighting differences and similarities. - Page 6, lines 57-58. I would specify how many focus groups have been carried out (not only the number of people taking part in them). - FIGURE 1 (conceptual framework): I believe that making the Figure 1 even more self-explanatory would add value to the paper. Considering its relevance for the structuring of the analysis and results, I would include some brief key terms in the Figure that would briefly take the main ideas describe in the text (page 8, lines 55-59; page 9, lines 3-21).
---

	- Page 9, lines 25-28: please, clarify why research results are not disseminated to participants, but instead to Government stakeholders. RESULTS: Even if references to interviews and focus groups statements are used in this section to support results, it would also be interesting to make references to the observational field work, if possible and useful. - Page 12, line 6: Please, modify and use same pattern to use same style and information always, when referencing information from interviews or focus groups. DISCUSSION: - Please, add additional reflections on the applicability and transferability of the results and conclusions to other settings, while exploring and proposing on possible implications for decision-making and healthcare delivery processes in India in the final paragraphs. REFERENCES: I would advise reviewing and completing those references with the citation date in the references that are taken from Internet and include a direct link. I would be happy to revise the version with all modifications once you are able to consider them. Good luck. Regards,
--	--

VERSION 1 – AUTHOR RESPONSE

Reviewer 1

Thank you for this paper on such an important topic : and presumably even more relevant given the current surge of COVID19 in India.

Reviewer 1, comment 1:

Overall I think this paper is interesting, however there seems to be two competing aims.

1.Experiences of FLHW with digital health records in 2 Indian states

2.The issues surrounding collection of mobile numbers to enable and deliver timely maternity care
I am not sure why they are mixed in this paper? It makes the paper confusing, with an emphasis on mobile phone numbers making a clear clinical information system focus difficult. We do not really explore the issues of clinical data collection in depth (mobile phone numbers are not clinical information). Then there is significant discussion around the delivery of maternity financial benefits and telehealth maternity education and advice (again not a usually function of clinical information systems).

I suggest two papers:

1.exploring the experiences of clinical information systems in the 2 states

2.the experiences of delivery digital enabled maternity care in the 2 states

As currently presented, it is difficult to draw coherent conclusions on either from this work. Thank you for addressing this important topic in a low resource environment.

Authors' response: Thank you for this comment and for identifying this confusion. We have sought to

clarify in the paper that our interest in understanding problems with mobile phone numbers led us to examine FLHW experiences overall with digital health records, because it was clear that these issues were inter-related.

The impetus for this work was to understand why so many attempts to send Kilkari phone calls to beneficiaries were not delivered and why so many beneficiaries received Kilkari quite late into their pregnancies or after childbirth, thus missing a large percentage of the program's content. To receive Kilkari, women's mobile phone numbers had to be accurately collected and digitized in government pregnancy registries. So we sought to understand where errors might be introduced, whether new numbers were updated in the system, and why mobile numbers were being entered late into a woman's pregnancy or after the child's birth. We quickly realized that mobile phone numbers are one field of data within many, and that the reasons for inaccuracies, out of date information, and late digitization are intertwined with collecting and digitizing other types of clinical and non-clinical data. Accuracy and timeliness of collecting and digitizing data depends on and the frontline health workers' experiences and work environments. Thus to identify contributors to inaccurate phone numbers and late creation of records we had to learn about (and explain in the paper) many other aspects of FLHW data collection and relationships.

We have revised the paper, and particularly the introduction, to more clearly explain that our primary interest was mobile numbers but that to understand phone numbers we had to discuss and report on the broader data system.

Reviewer 1, comment 2:

p14 line 13 please correct spelling error "because they ould not rely on the ASHAs to complete any paper forms"

Authors' response: Thank you, we have corrected this error.

Reviewer 1, comment 3:

p17. This sentence seems to contradict itself: please review. "We found that frontline workers and their supervisors prioritized the accuracy of core health information and of data elements that enabled pregnant clients to receive government financial entitlements (e.g., bank account information, identification numbers) over timeliness or the accuracy of beneficiary mobile phone numbers"

Authors' response: Thank you. We have edited the sentence as follows: "We found that frontline workers and their supervisors prioritized the accuracy of data elements that enabled pregnant clients to receive government financial entitlements (e.g., bank account information, identification numbers) over mobile phone numbers."

Reviewer 1, comment 4:

p18 there is an overwhelming focus on mobile phone numbers. I suspect perhaps I don't understand the context to understand why the mobile numbers are so important. Mobile numbers are core administrative data and are not really part of the clinical record. Mobile number collection can be part of a patient administrative system (PAS) rather than an electronic medical or health record. (EMR/ EHR)

Authors' response: Thank you for this observation. As discussed above, we have now clarified why we were so interested in accurate, up-to-date and timely registration of phone numbers, i.e., so that women could receive Kilkari mHealth messages throughout their pregnancy and child's first year of life. We hope now that the focus on phone numbers is logical. We also clarified in the introduction that phone numbers are embedded in the broader data collection, digitization and use context.

Reviewer 2

Dear authors,

The submitted manuscript is interesting and gives a valuable new perspective of the barriers to the accurate, complete and timely creation of health information systems used for maternal, child, and reproductive health in India.

Although its appropriateness for its publication, I suggest some points have to be reviewed and modify in the manuscript before its acceptance.

Please, be aware that page reference in the following lines is taken from the top side of the pages.

Reviewer 2, comment 1:

OVERALL REMARK:

- I would include a dictionary (Annex 1) with the most repeated and relevant acronyms in the manuscript to facilitate comprehension of all included terms. Please, consider also to add the complete description of all acronyms at least once in the manuscript (the first time they appear).

Authors' response: This is a very good idea. We have created this dictionary.

METHODS:

Reviewer 2, comment 2:

- Page 6, lines 46-50: As it is stated "the states provide a natural comparative study". In this sense, I would include this piece of information in the last paragraph of the Introduction section, where the objectives of the paper are presented.

Furthermore, if the comparison is allowed and making a comparison is also an objective of the research, it would also be interesting to present data in the results section in a comparative way, highlighting differences and similarities.

Authors' response: Thank you. We see that we have been imprecise with our language. The study did not set out to compare these two states as a primary goal. Instead, the study set out to understand accuracy and completeness of data in the government registry in India, and selected two states that could provide variation in terms of their pregnancy registries. We have edited the language to now read: "The states thus enable us to examine data systems in a more typical case (Madhya Pradesh), which, like most large Indian states, recently moved from MCTS to RCH, and an outlier case (Rajasthan) which has retained a tailor-made electronic record system since the beginning of digitization."

Reviewer 2, comment 3:

- Page 6, lines 57-58. I would specify how many focus groups have been carried out (not only the number of people taking part in them).

Authors' response: In fact we refer to the number of focus groups, not the number of people who took part. We have clarified this in table 3 by specifically stating "Number of FGDs"

Reviewer 2, comment 4:

- FIGURE 1 (conceptual framework): I believe that making the Figure 1 even more self-explanatory would add value to the paper. Considering its relevance for the structuring of the analysis and results, I would include some brief key terms in the Figure that would briefly take the main ideas describe in the text (page 8, lines 55-59; page 9, lines 3-21).

Authors' response: Thank you for this suggestion. We have re-worked our framework to add additional explanations.

Reviewer 2, comment 5:

- Page 9, lines 25-28: please, clarify why research results are not disseminated to

participants, but instead to Government stakeholders.

Authors' response: Our findings were of specific interest to the government policy makers, who were seeking insight into improving the quality of mobile phone numbers in their MCTS/RCH registries. In light of constraints in time and resources, we focused on disseminating findings to administrators in government. We have indicated this in the appropriate section, noting "Results were disseminated to Government of India stakeholders but not to research participants due to the policy-level implications of our findings."

Reviewer 2, comment 6:

RESULTS:

Even if references to interviews and focus groups statements are used in this section to support results, it would also be interesting to make references to the observational field work, if possible and useful.

Authors' response: Thank you. We integrated many findings from our observations but were forced to remove some due to work length constraints. We have added one back in, on observing the non-universality of checking the accuracy of mobile numbers through giving a missed call during FLHW-beneficiary interactions.

Reviewer 2, comment 7:

- Page 12, line 6: Please, modify and use same pattern to use same style and information always, when referencing information from interviews or focus groups.

Authors' response: Thank you, we have ensured that all references to interviews and focus groups now use the same style and present the same information.

Reviewer 2, comment 8:

DISCUSSION:

- Please, add additional reflections on the applicability and transferability of the results and conclusions to other settings, while exploring and proposing on possible implications for decision-making and healthcare delivery processes in India in the final paragraphs.

Authors' response: Thank you. We have edited the discussion to incorporate additional clear implications for decision makers in India.

Addition to end of paragraph 3: These processes can inform beneficiaries and frontline providers about the benefits of ensuring accurate mobile numbers are registered in the digital system (including that beneficiaries can be subscribed to mHealth programs such as Kilkari), potentially increasing both parties' motivation to keep the registry up-to-date.

Addition to end of paragraph 4: Health workers require training on the value of maintaining an up-to-date mobile number in their data registries, so that they are motivated to ensure these numbers are correct. Furthermore, in light of research findings that "SIM churn" (changing SIM cards, and as a result getting a new mobile number) is as high as 44% per year among rural families,¹⁰ frontline workers need to regularly confirm beneficiary mobile numbers and have access to user-friendly mechanisms to update numbers whenever they change.

We have also added the following paragraph at the end of the discussion on applicability to other settings:

Three core determinants of accurate, complete and timely administrative data emerged from our research and are likely applicable to health registries for pregnant and postpartum women in other lower resource settings. First, beneficiaries who trust their health workers and the government more broadly are willing to provide accurate data.^{25,26} Second, data entry systems that are easy to use, appropriate to the local context, and useful to the frontline worker facilitate data accuracy and

timeliness.27,28 And third, supportive supervision and adequate resources are vital to building frontline health worker skills and knowledge across clinical and administrative functions.29–31

Reviewer 2, comment 9:

REFERENCES: I would advise reviewing and completing those references with the citation date in the references that are taken from Internet and include a direct link.

Authors' response: Thank you. We have reviewed the references.

Reviewer 2, comment 10:

I would be happy to revise the version with all modifications once you are able to consider them.

Good luck.

Authors' response: Thank you very much.

VERSION 2 – REVIEW

REVIEWER	Sullivan, Clair The University of Queensland Faculty of Medicine and Biomedical Sciences, Centre for Health Services Research
REVIEW RETURNED	25-Dec-2021

GENERAL COMMENTS	my comments from last review have been addressed many thanks
--

REVIEWER	Urtaran-Laresgoiti, Maider University of Deusto, Deusto Business School Health
REVIEW RETURNED	27-Dec-2021

GENERAL COMMENTS	Dear authors, I have noticed a big step toward a better-structured version of the paper, and I believe changes in the manuscript have improve the quality of it considerably. As I stated in my last letter to you, I consider the research being valuable, for managers and policy makers specifically, in their responsibility to enhance quality of health care. Although improvements are notorious, I suggest authors to review some points before publication. Below, I detail some suggestions and questions for you to consider when reviewing the manuscript. METHODS:  - Table 3: Respondents sample: Please, specify if possible number of participants in each FGD. It would also be interesting to refer to the homogeneity or heterogeneity of the sample, its representativeness, etc. - Page 6. Line 60. Is all the data collection (the 59 IDIs, 12 FGDs) and entry process carried on in one month (September 2018)? RESULTS: Even if some reference to observational fieldwork has been introduced, I would advise to review and support the section with more, if you consider useful and it is possible. It would also be relevant to include an equity perspective and provide a commentary on the differences in perceptions or barriers that may arise between women from different ethnic, socioeconomic or educational levels (if data available). In addition
--

	to it, a brief mentioning in the discussion section with respect to the implications that these differences may arise. DISCUSSION: Please, add additional reflections on the applicability and transferability of the results and conclusions to other settings. Once more, I would be happy to revise the version with all modifications once you are able to consider them. Good luck. Regards,
--	--

VERSION 2 – AUTHOR RESPONSE

Request 2: Please ensure that you have fully discussed the methodological limitations of the study in the discussion section of the main text.

Response 2: We have added the following paragraph:

This study has some limitations, which point towards areas for future research. The information presented here was either gathered through interviews and focus groups and thus self-reported, or was gathered through observation, wherein data entry operators were aware that their work was being observed. Thus respondents may have minimized or hidden unauthorized behavior, such as shortcuts to speed up data entry or unsanctioned task-shifting. Future quantitative research could more objectively measure data delays (e.g., time elapsed from first antenatal interaction with an actor from the government health system to the creation of a digital record) and data accuracy (e.g., systematically validating key data fields, including mobile phone numbers, for a sample of records). For this study, we spoke to pregnant and post-partum women with mobile phones who recently interacted with health system actors: this sample enabled us to understand reasons for delays and inaccuracies among beneficiaries whose data should have been easy to digitize. Future research among women without mobile phones and women who interacted with the government health system many months after childbirth would triangulate understanding about the data sharing practices of these additional populations.

Request 3: Please include a copy of the interview guide as a supplementary file or a link to where readers can access it.

Response 3: We have now created a supplementary files with the IDI guide with frontline workers and the FGD guide for beneficiaries.

Comments from reviewer 2:

METHODS:

Comment 1: - Table 3: Respondents sample: Please, specify if possible number of participants in each FGD. It would also be interesting to refer to the homogeneity or heterogeneity of the sample, its representativeness, etc.

Response 1: We have added the following information:

“FGDs ranged from four to 10 participants (mean 7.6 participants). The focus group compositions broadly reflected local demographics. They included women with a wide range of education levels

(from no education to master's degrees), castes (most included a mix of women from marginalized schedule caste and schedule tribal groups as well as women from "other backwards castes" and general caste groups), and religions (three included some Hindu and some Muslim women, while the remainder were all Hindu). Most women were homemakers, while a sizable minority worked as agricultural farmers and labourers, and also included students, tailors, shopkeepers and bangle/jewelry saleswomen."

Comment 2: Page 6. Line 60. Is all the data collection (the 59 IDIs, 12 FGDs) and entry process carried on in one month (September 2018)?

Response 2: We have noted that data was collected in Sept and Oct 2018.

RESULTS:

Comment 3: Even if some reference to observational fieldwork has been introduced, I would advise to review and support the section with more, if you consider useful and it is possible.

Response 3: Thank you for this suggestion. Given our wordcount and the robustness of the existing evidence presented to support our findings, we have reluctantly decided that integrating additional observation data. We feel that upon reviewing the observation data, adding more observations would not substantially strengthen the narrative presented and is thus not worth the added words.

Comment 4: It would also be relevant to include an equity perspective and provide a commentary on the differences in perceptions or barriers that may arise between women from different ethnic, socioeconomic or educational levels (if data available). In addition to it, a brief mentioning in the discussion section with respect to the implications that these differences may arise.

Response 4: This is a very good suggestion. Our data from beneficiaries was all FGD data and no clear patterns emerged along education, caste or religion lines. However, you are absolutely correct that differences surely exist. We have thus added a call for future research to delve into this equity issue more deeply in the discussion section: "Furthermore, research with additional beneficiaries could provide richer insight into equity dimensions related to different caste, religion, socioeconomic status or educational levels. Such insights would best be gleaned through targeted in-depth interviews with specific marginalized populations and FGDs that separated women by caste, religion, socioeconomic status or educational levels."

DISCUSSION:

Comment 5: Please, add additional reflections on the applicability and transferability of the results and conclusions to other settings.

Response 5: We earlier added the following reflections on applicability and transferability:

Three core determinants of accurate, complete and timely administrative data emerged from our research and are likely applicable to health registries for pregnant and postpartum women in other lower resource settings. First, beneficiaries who trust their health workers and the government more broadly are willing to provide accurate data.^{25,26} Second, data entry systems that are easy to use, appropriate to the local context, and useful to the frontline worker facilitate data accuracy and timeliness.^{8,27} And third, supportive supervision and adequate resources are vital to building frontline health worker skills and knowledge across clinical and administrative functions.^{28–30}

We feel it may be over-reaching to expand statements of applicability and transferability beyond these three points.

Sincerely,
Kerry Scott on behalf of the co-authors